# Electron-flux infrared response to varying π-bond topology in charged aromatic monomers

Héctor Álvaro Galué[1], Jos Oomens[1,2], Wybren Jan Buma[1] & Britta Redlich[2]

The interaction of delocalized π-electrons with molecular vibrations is key to charge transport processes in π-conjugated organic materials based on aromatic monomers. Yet the role that specific aromatic motifs play on charge transfer is poorly understood. Here we show that the molecular edge topology in charged catacondensed aromatic hydrocarbons influences the Herzberg-Teller coupling of π-electrons with molecular vibrations. To this end, we probe the radical cations of picene and pentacene with benchmark armchair- and zigzag-edges using infrared multiple-photon dissociation action spectroscopy and interpret the recorded spectra via quantum-chemical calculations. We demonstrate that infrared bands preserve information on the dipolar π-electron-flux mode enhancement, which is governed by the dynamical evolution of vibronically mixed and correlated one-electron configuration states. Our results reveal that in picene a stronger charge π-flux is generated than in pentacene, which could justify the differences of electronic properties of armchair- versus zigzag-type families of technologically relevant organic molecules.

[1] Van't Hoff Institute for Molecular Sciences, University of Amsterdam, Science Park 904, 1098XH Amsterdam, The Netherlands. [2] Radboud University, Institute for Molecules and Materials, FELIX Laboratory, Toernooiveld 7, 6525ED Nijmegen, The Netherlands. Correspondence and requests for materials should be addressed to H.A.G. (email: h.alvarogalue@gmail.com) or to B.R. (email: felix@science.ru.nl).

A challenge in the innovation of molecular organic electronics is to understand the fundamental physical principles controlling charge transport[1,2]. Significant efforts have focused on acene[3] aromatic hydrocarbons consisting of fused benzenoid rings arranged in centrosymmetric linear structures. Acenes are key monomeric building blocks for designing prototypical organic solids as the number of rings defines the electronic band structure[3] via the extent of molecular $\pi$-conjugation or delocalization (in which overlapping $p_z$ atomic orbitals interconnect electrons across rings). Another approach to organic electronics is offered by phenacenes[4,5] which are non-centrosymmetric versions of acenes consisting of rings fused in angular-oriented structures. Illustrative examples in the case of molecular crystal solids based on acenes and phenacenes show diverse electronic properties[3–9] ranging from semiconducting to metallic and even superconducting. In particular, pentacene of five linearly fused rings, is a common active compound used in field-effect semiconductors[6]. The semiconducting phenacene counterpart of pentacene is picene, which exhibits important differences in charge mobility and chemical stability[4,7,8,10]. A more intriguing distinction between the two monomers is the reported superconductivity of metal-doped picene solids, which is absent in pentacene analogues[9,11–13]. While the crystal configuration (for example, lattice, doping, chemical-group functionalization) and operational conditions govern the conductivity in the above examples, one can expect that the intrinsic picene and pentacene molecular structures play a decisive role as well.

Presently, a molecular-level picture reconciling the differences in electronic properties is lacking. Yet, a recognizable influential factor on these properties is the vibronic coupling of molecular vibrations with $\pi$-electron molecular orbitals[12,14–18] inherent to the $\pi$-bond-edge topology[19]. To gain insight in the role of this topology in picene and pentacene vibronic behaviours, we probe here their monomeric structures in the positive charge state (picene$^+$, pentacene$^+$) using infrared multiple-photon dissociation action spectroscopy[20,21]. This spin-doublet cationic state, featuring an unpaired electron in the highest-occupied molecular orbital, vibronically couples with electronic states associated with excitation to low-lying unoccupied molecular orbitals of the proper symmetry. In our experiments we isolate gas-phase cations in an ion trap and probe them with infrared photons from the Dutch free-electron laser (FEL) for infrared experiments. By means of resonant multiple-photon vibrational excitation, we record photodissociation spectra as we tune the FEL photon energy. This high-sensitivity technique helps us circumvent the issue of undetectable direct absorptions of the low-density ion samples that result from electrostatic repulsion. Although quantification of action spectra can be non-trivial due to the multiple-photon dissociation dynamics[20–22], the central thesis here is the isomeric correspondence between picene and pentacene in which multiple-photon dissociation channels have spectral responses affected by alike intrinsic kinetics. Thus, in this case, action spectra of two species can be compared quantitatively provided that the spectra are measured under similar experimental settings.

The spectra of both cationic systems show substantial infrared activity in the 1,100–1,600 cm$^{-1}$ range, which we ascribe to electronic density oscillations during antisymmetric C=C stretch vibrational excitation. The driving vibronic mechanism[23–26] arises in the molecular dipole moment ($\boldsymbol{\mu}$) derivative along the nuclear displacement normal-coordinate $Q_k$ of the infrared intensity equation (ref. 27) $I_k = (8\pi^3 N v_k/3hc)|\langle V_1|(\partial\boldsymbol{\mu}/\partial Q_k)Q_k|V_0\rangle|^2$, where the vibrational wavefunctions $V_0$ and $V_1$ characterize the fundamental harmonic dipole transition (1←0) of $v_k$th mode. The second term of the molecular dipole $\boldsymbol{\mu} = \varrho_N(R) + \langle\Psi_g(r;R)|$

$(-e\mathbf{r})|\Psi_g(r;R)\rangle$ is the non-classical dipolar part of ground-state electrons described by the Born–Oppenheimer wavefunction $\Psi_g$, $r$ and $R$ being the electronic and nuclear coordinates, and $\varrho_N$ being the nuclei dipole. By equating a first-order expansion of $\Psi_g$ with respect to nuclear normal coordinates $Q$ (Supplementary Note 1) into $\boldsymbol{\mu}$ we write $\partial\boldsymbol{\mu}/\partial Q_k$ as a sum of two sources of infrared activity[24,28]

$$\frac{\partial\boldsymbol{\mu}}{\partial Q_k} = \frac{\partial}{\partial Q_k}[\varrho_N + \langle\Psi_o|-e\mathbf{r}|\Psi_o\rangle] + 2\sum_{\xi=i}\frac{\langle\Psi_o|(\partial H/\partial Q_k)_0|\Psi_i\rangle}{E_i - E_o}\langle\Psi_o|\mathbf{r}|\Psi_i\rangle, \quad (1)$$

where single (one-electron) configuration wavefunctions $\Psi_{o/i}$ (evaluated at equilibrium nuclear positions) and energies $E_{o/i}$ correspond to ground (o) and excited (i) adiabatic states ($\xi$). The matrix elements $\langle\Psi_o|(\partial H/\partial Q_k)_0|\Psi_i\rangle$ and $\langle\Psi_o|\mathbf{r}|\Psi_i\rangle$ are the Herzberg–Teller vibronic coupling strength and electronic transition, respectively, and $H$ is the electronic Hamiltonian. The first static-charge term arises from oscillating nuclei and nuclei-fixed electrons, while the second charge-flux[29–31] term of oscillating non-fixed electrons is the vibronically active contribution of low-lying electronic excited states mixing into the ground state.

We show that the vibronic contribution to the picene$^+$ infrared spectrum can be twice as large as in the pentacene$^+$ spectrum, which is attributed to stronger dipole $\pi$-electron fluxes in the former system. Quantum-chemical calculations support this conclusion but also reveal that electronic correlation is essential to describe the intense C=C stretch $\pi$-flux modes of picene$^+$. Whereas the multiple-photon dynamics impedes extracting absolute magnitudes of the underlying $\pi$-fluxes in picene$^+$ and pentacene$^+$, by virtue of comparing their multiple-photon dissociation yields we are able to discern molecular charge-flux effects (driven by vibronic coupling) on the resultant action spectra. We show that within the used FEL settings, the recorded action bands scale quasi-linearly with FEL average power and are satisfactorily described under the harmonic approximation as corroborated by quantum-chemical theory (apart from the anharmonic band broadenings inherent to the multiple-photon excitation process). We assert that the ability of picene to enable a significant dipolar $\pi$-flux charge separation, as manifested in the infrared action bands, is a general characteristic of aromatic motifs with armchair-edge topology. The fundamental distinction found here between picene and pentacene provides a dynamical charge-flux—structure relation useful to rationalize charge transport phenomena in $\pi$-conjugated organic materials built from aromatic structures.

## Results

**Molecular structures.** We first examine the structural edge topology[32] (Fig. 1a) in terms of $\pi$-electron delocalization. With zigzag edges, pentacene has only one resonant ring sextet of interconnected $\pi$-electrons[33], while the other 16 non-sextet $\pi$-electrons tend to stay within bonds. Conversely, the picene armchair edges enable three $\pi$-sextets to resonate into adjacent rings[33], and this fact fosters an aromatic system in which delocalization extends over the entire structure. This justifies the higher stability of neutral picene over pentacene by 0.68 eV calculated by density functional theory (B3LYP/6-311G**). The degree of $\pi$-delocalization is then inherently different in both systems as reflected in the $\pi$-electron spin-orbital wavefunctions. Figure 1b shows the energetic orderings with different spacing of frontier molecular $\pi$ spin-orbitals. The orderings define the electronic configurations of cationic spin-doublet ground states ($D_o$) $^2B_1$ and $^2B_{2g}$ of picene$^+$ and pentacene$^+$, respectively, and

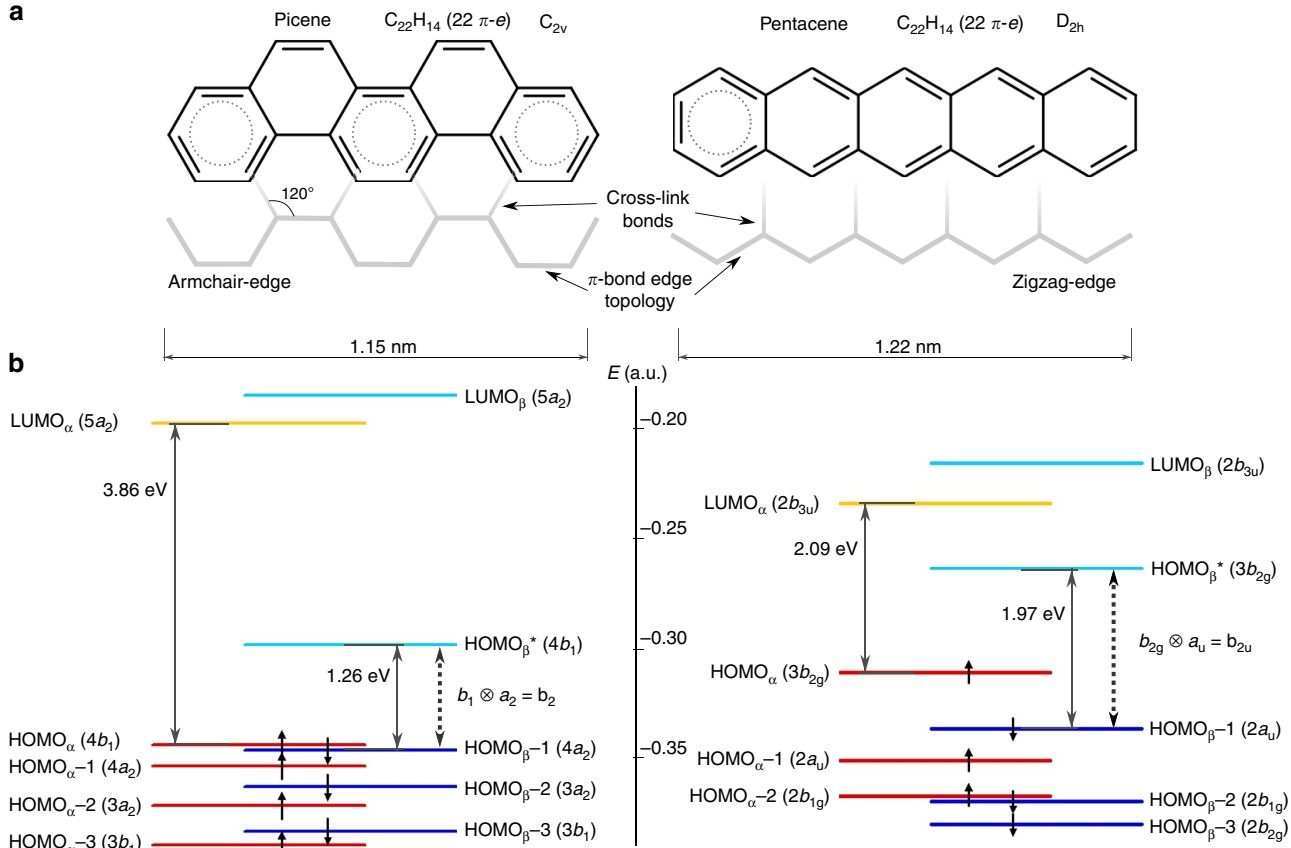

**Figure 1 | Investigated monomers and frontier π spin-orbital energy levels.** (**a**) Chemical structures of picene ($D_{2h}$ symmetry) and pentacene ($C_{2v}$ symmetry) featuring delocalized sextet rings of six π-electrons (dotted inner circles). (**b**) Energy-level configurations of molecular π spin-orbitals in the cationic spin-doublet ground state (B3LYP/6-311G**). The energy gaps between spin-orbitals are governed by cross-link bond electrostatic interactions[19]. The vertical cross-link bonding between two zigzag edges causes out-of-phase atomic-orbital combinations[32] that tend to disperse frontier spin-orbitals, while the interaction between armchair edges takes place through inner *ca.* 120° angle-oriented cross-linkage of in-phase character, resulting in spin-orbitals that bear close proximity or even accidental degeneracy. Dotted arrows show main vibronically active $π → π^\star$ orbital excitations along $b_2$ (picene$^+$) and $b_{2u}$ (pentacene$^+$) symmetry modes. LUMO stands for lowest-unoccupied molecular orbital and 1 energy atomic unit (a.u.) is 27.21 eV.

of low-lying excited states accessible via $π → π^\star$ excitations (Supplementary Tables 1 and 2). Experimentally, we produce pentacene and picene radical cations ($C_{22}H_{14}^{+•}$) by non-resonant two-photon ionization[34] of the neutral precursors at 193 nm (Fig. 2).

**Infrared action spectra.** A FEL macropulse resonantly energizes the initially thermal population (Supplementary Fig. 1) of mass-isolated $C_{22}H_{14}^{+•}$ parent ions (picene$^+$ or pentacene$^+$) via absorption of several tens of infrared photons resulting in dissociation to product ions $C_{20}H_{12}^{+•}$ and $C_{18}H_{10}^{+•}$ (Fig. 2b,d). Note that the fast intramolecular vibrational energy redistribution ($IVR \gg 10^9 s^{-1}$) among normal modes ensures statistical allocation of the energy of each absorbed photon prior to dissociation[35], as well as the resonant absorption at fundamental transitions[36,37]. Under the FEL settings of our experiments, resonant excitation by the FEL induces competing dissociation kinetics of $C_2H_2$ (26 u) versus $C_4H_4$ (52 u) loss channels (Fig. 2b). The signal of the dominant $C_2H_2$-loss channel typically comprises only 2 and 7% of picene$^+$ and pentacene$^+$ thus obviating the occurrence of dissociation yield band saturations around the on-resonance region (that is, band *b*). The higher 7% in pentacene$^+$ mainly arises from a relatively lower parent signal for this particular experimental run.

The final action spectra plotted along wavelength or frequency (Figs 2c and 3) result from averaging various dissociation yield functions $β(λ)$ retrieved from the product ion signals recorded along FEL photon energy (Methods). The final average of both systems is corrected for FEL power variation and normalized to 1. The two spectra include a low-energy range extending down to 400 cm$^{-1}$. The power correction normalizes the band intensities from different spectral scans as measured in independent FEL-ion-trap experimental sessions. The precision in our measurements is discussed below for a data set sample of single scans (Supplementary Fig. 2). Apart from the large random noise component and slight variations in band broadening, each single spectrum in the sample exhibits similar infrared absorption features as confirmed by the partial spectra averages featuring reduced random noise level (Supplementary Figs 3 and 4 with band characterizations in Supplementary Tables 3–6).

The statistical random noise in each one-scan spectrum produces somewhat diverse baselines upon dividing by the FEL power curve (hence, averaging usually precedes the power correction). Also, the high noise in each spectrum impedes performing the spectral deconvolution curve fitting since the parameter initialization (based on the initial guess of band peak positions) fails. We thus obtain the frequency ($\tilde{v}_{exp}$) and intensity ($β$) band values manually. Despite the inaccuracy added by this human factor, the statistics from the data sample provides a notion of the precision between measurements. The $\tilde{v}_{exp}$ (and $β$) band characterizations are $a_r$: $1,567 ± 19$ cm$^{-1}$ ($0.50 ± 0.10$), $a$: $1,512 ± 17$ cm$^{-1}$ ($0.73 ± 0.09$), $b_r$: $1,338 ± 10$ cm$^{-1}$ ($0.61 ± 0.05$),

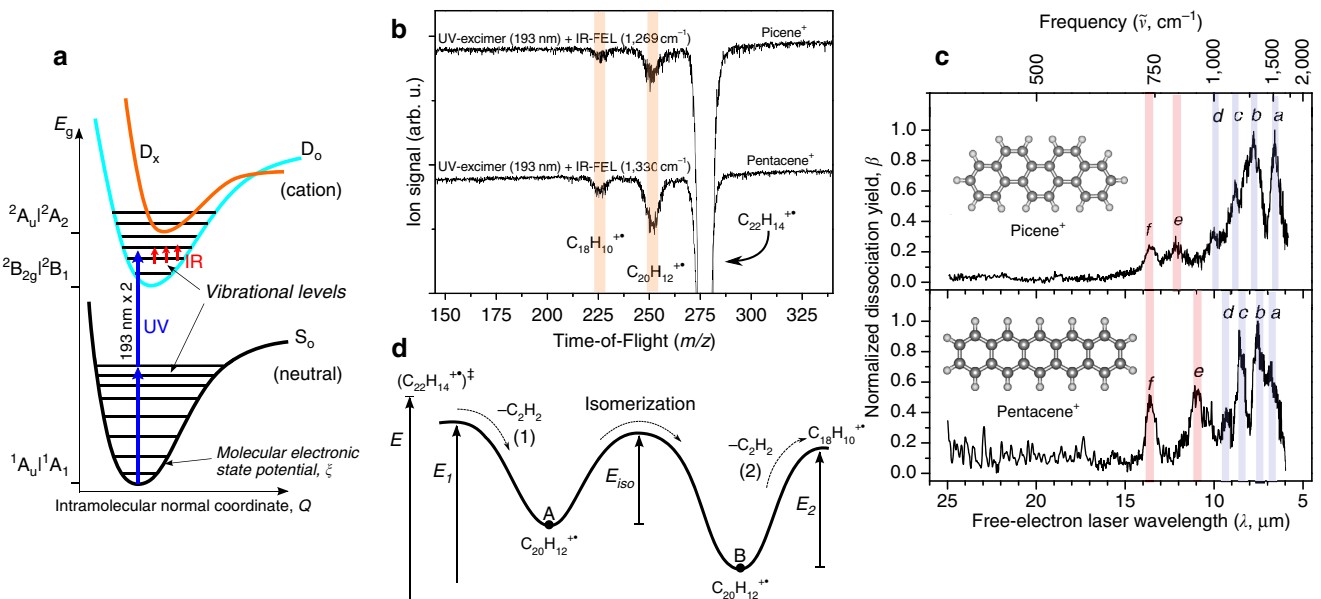

**Figure 2 | Laser photoexcitation dynamics and multiple-photon dissociation action spectra.** (**a**) Schematic of adiabatic electronic potentials involved in ultraviolet and infrared photo-excitations leading respectively to charged $D_o$-state aromatic monomers and vibrational infrared action spectra. The spin-doublet excited state $D_x$ closest to the ground state $D_o$ is defined by the singly excited configuration $\Psi_i(4a_2 \rightarrow 4b_1)$ for picene$^+$ and $\Psi_i(2a_u \rightarrow 3b_{2g})$ for pentacene$^+$ (Supplementary Tables 1 and 2). The y-axis equilibrium electronic energy ($E_g$) shows the corresponding ground-state molecular terms. (**b**) Typical mass analysis upon ultraviolet photoionization and infrared FEL macropulse irradiation. The ionization energies of picene and pentacene are 7.5 and 6.6 eV, respectively[19], and the excess of vibrational excitation determined by the Franck–Condon overlaps of neutral- and cationic-state vibrational wavefunctions partially evolves as $C_2H_2$-loss dissociation (Methods). (**c**) Infrared action spectra ($\beta(\lambda)$) retrieved from $C_{20}H_{12}^+{}^\bullet$ ($m/z$ 252) and $C_{18}H_{10}^+{}^\bullet$ ($m/z$ 226) ion signals recorded continuously along tuned FEL photon energy (action bands characterized in Table 1). (**d**) Proposed scheme of multiple-photon induced dissociation kinetics along the potential energy reaction profile (reaction coordinate from left to right) connecting the reactant $C_{22}H_{14}^+{}^\bullet$ parent ion (as transition state ‡)[35] with $C_2H_2$-loss product ion (A), the intermediate isomerization[34] product ion (B), and the $C_4H_4$-loss product ion ($C_{18}H_{10}^+{}^\bullet$). The energy barriers $E_1$, $E_{iso}$, and $E_2$ correspond to first $C_2H_2$-loss (1), isomerization and second $C_2H_2$-loss (2) reactions. Figure 6 confirms that the two product ions (both exhibit same spectral features) arise from the same parent; hence, the possible event of sequential photon absorption by the intermediate B structure does not influence the spectra.

*b*: 1,281 ± 9 cm$^{-1}$ (0.75 ± 0.05), $b_l$: 1,212 ± 8 cm$^{-1}$ (0.62 ± 0.04), and *c*: 1,141 ± 10 cm$^{-1}$ (0.56 ± 0.07) for *picene*$^+$; and *a*: 1,448 ± 16 cm$^{-1}$ (0.45 ± 0.06), *b*: 1,313 ± 9 cm$^{-1}$ (0.78 ± 0.09), *c*: 1,176 ± 7 cm$^{-1}$ (0.77 ± 0.09), *e*: 911 ± 7 cm$^{-1}$ (0.47 ± 0.05), and *f*: 737 ± 4 cm$^{-1}$ (0.43 ± 0.04) for *pentacene*$^+$. These values are in fair agreement with the values determined via the more rigorous deconvolution procedure applied on sample averages (B2B3 and A for picene$^+$; D1D3D5 and D6D7 for pentacene$^+$) featuring a reduced noise (Supplementary Tables 3–6), and with the values of the full data set average (Fig. 3). We establish that the $\tilde{\nu}_{exp}$ and $\beta$ band fluctuations are due to typical random changes during experiments rather than to systematic instrumental biases or uncharacterized molecular processes during ion trapping. Finally, note that averaging over the full data set reduces the noise component more than when averaging over partial data sets. Thus, the larger random errors in the partial spectra are reflected in the spectral curve fittings characterizing the measured bands. For picene$^+$ *b* band, the $\tilde{\nu}_{exp}$ and $\beta$ average values between B2B3 and A data samples are 1,279.5 cm$^{-1}$ and 0.87. These values represent a deviation of 0.20 and 6.5% relative to final values (Table 1) and accounts in part for the higher random error of partial spectra. From D1D3D5 and D6D7 data samples of pentacene$^+$, the $\tilde{\nu}_{exp}$ (and $\beta$) average is 1,317.5 cm$^{-1}$ (0.89) and represents a random error deviation of 0.11% (5.3%).

**Harmonic analysis.** Although the multiple-photon excitation relies on the anharmonic character of IVR coupled modes, which could induce noticeable nonlinear effects[37], there are examples such as the naphthyl$^+$ action spectrum[38,39] confirming the

quasi-resonant harmonic description of adiabatic molecular potentials probed by multiple-photon action spectroscopy[37,40]. Here, the bands observed in the action spectra are summarized in Table 1 with their spectral deconvolutions and harmonic mode assignments (from Supplementary Tables 7–16) based on the B3LYP vibrational analysis of Fig. 3. Both cations feature a very intense high-energy 1,600–1,100 cm$^{-1}$ region of *a*, *b* and *c* bands, and a relatively weak mid-energy 1,100–700 cm$^{-1}$ region of *d*, *e* and *f* bands. The band decompositions in Fig. 3a,f reveal the extent to which individual normal modes contribute to action bands. The mean value of absolute shifts between action and harmonic band frequencies is 1.43 ± 0.9% in picene$^+$ and 1.19 ± 0.79% in pentacene$^+$. The larger total shift in picene$^+$ reflects the contribution of the $C_4H_4$-loss dissociation channel (larger than in pentacene$^+$) for which high activation energy brings larger anharmonic shifts[20] (*vide infra*).

Despite the higher symmetry in pentacene$^+$, its action spectrum reveals more spectral congestion than picene$^+$ action spectrum as confirmed by its bigger number of fitted peak components (Fig. 3a,f). The high-energy region of $b_2$–symmetry modes in picene$^+$ and $b_{2u}$–symmetry modes in pentacene$^+$ comprises, respectively, 92% and 88% of the total infrared activity (Supplementary Table 17). Generally, $b_2$ and $b_{2u}$ modes involve $C=C$ stretching motions antisymmetric with respect to the molecular *xz* plane, and are susceptible to vibronic couplings with low-lying $\pi$-orbitals. The band decompositions show that the intense vibronically active modes $\nu_{78}$ and $\nu_{84}$ in picene$^+$ are responsible of bands *a* and *b* (see Fig. 3a,e), whereas the strongest mode $\nu_{85}$ in pentacene$^+$ generates band *b* (Fig. 3f,j). Next to

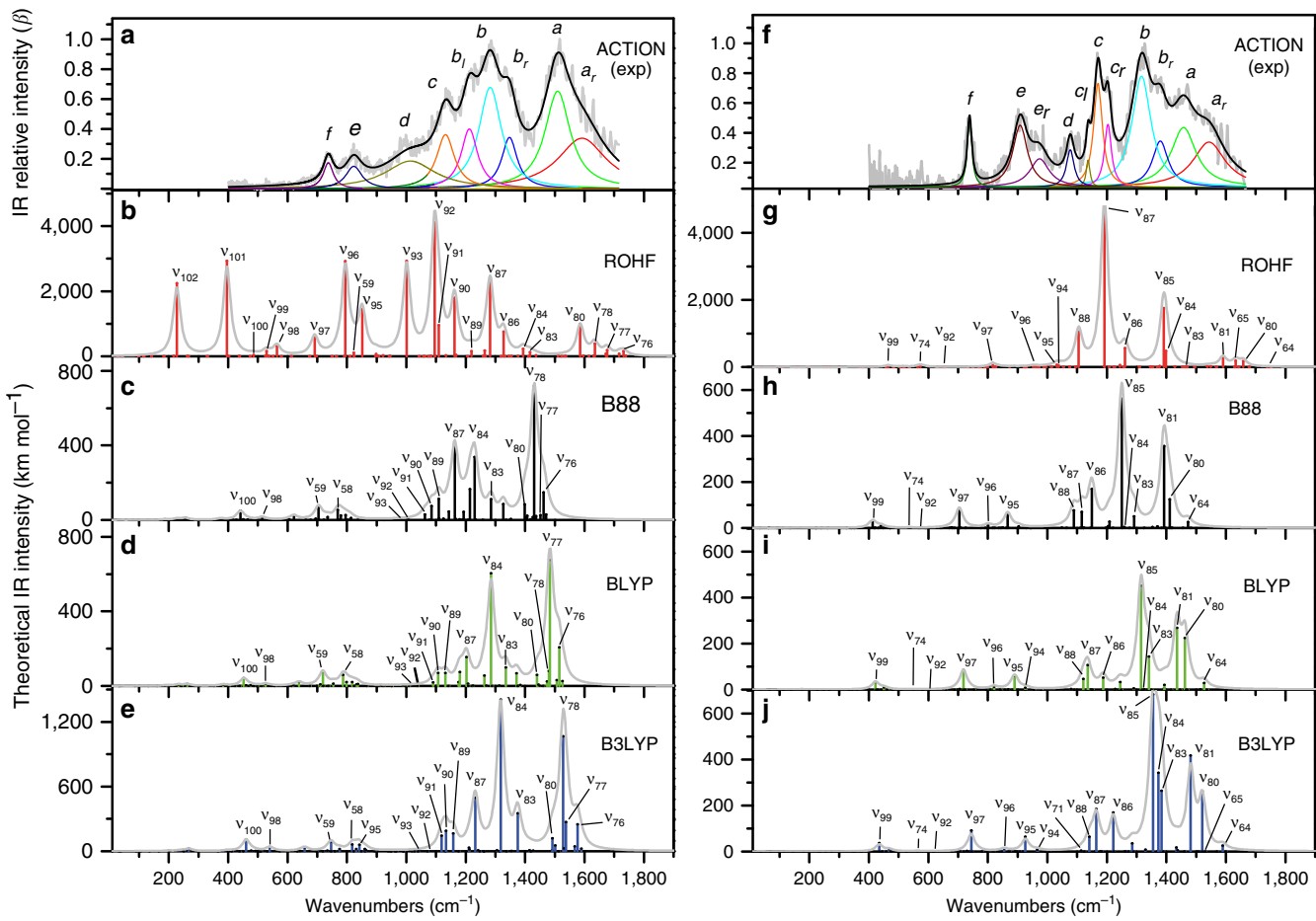

**Figure 3 | Harmonic mode analysis as a function of electronic correlation.** The resultant final action spectra (grey) converted to spatial frequency wavenumbers ($cm^{-1}$) of (**a**) picene$^+$ and (**f**) pentacene$^+$. The spectral deconvolution band fitting based on the Levenberg–Marquardt algorithm results, respectively, for picene$^+$ and pentacene$^+$, in 9 and 11 Lorentzian band peaks (each correspond to a lower case letter), which cumulative peak spectral curve is depicted in black. In Table 1 resultant peaks are summarized and sequentially numbered from $a_r$ to $f$ in correspondence with action band features. Theoretical linear harmonic spectra (generated by convoluting scaled normal modes with a Lorentzian 30 $cm^{-1}$ function) are based on ROHF (**b**, picene$^+$ and **g**, pentacene$^+$) and B88, BLYP and B3LYP density functionals (see Methods, mode assignments in Supplementary Tables 7–16 where B is B88) for **c–e,** picene$^+$ and **h–j**, pentacene$^+$.

band $b$, appears the band $b_r$ shoulder (decomposed into blue fitted peak) due to the photoexcitation at the lower-energy anharmonic part of the molecular potential[20] along modes $\nu_{83}/\nu_{84}$ in pentacene$^+$ and $\nu_{83}$ in picene$^+$. As seen in the spectra, and despite the band overlaps, during these anharmonic photo-absorptions the adjacent modes $\nu_{85}$ and $\nu_{84}$ clearly shift into FEL resonance, respectively, raising the dissociation yield in proportion to their (higher) infrared activity. In this case the final relative intensity of the $b$–$b_r$ band complex seems to reflect the strength of the most intense photoexcited mode in both action spectra.

The mid-energy band $f$, with identical peak position in both systems, results from photoexcitation at out-of-plane C–H bending modes $\nu_{59}$ in picene$^+$ and $\nu_{97}$ in pentacene$^+$. These modes involve in both systems bending motions of the four C–H bond oscillators at outer rings, thereby explaining the spectral equivalence. Band $e$, also with out-of-plane C–H bending character, arises from two and one C–H oscillators at the three inner rings of picene$^+$ ($\nu_{58}$) and pentacene$^+$ ($\nu_{95}$), respectively. In this case we observe a frequency difference between the two species of 10% that agrees with the prediction of 11%. This spectral shift translates into a 39% increase in restoring force (proportional to the local C–H bond charge[41]) revealing more electron density localization over C–H oscillators in pentacene$^+$.

As seen later, the relatively high intensities of $f$ and $e$ bands (see $\beta/I_{cal}$ ratios, Table 1) arise equally in picene$^+$ and pentacene$^+$ from an increased ion production through the $C_2H_2$-loss dissociation channel at these low excitation energies.

**Electronic correlation.** The dipolar charge redistribution along any vibration in both monomers is governed by the molecular electronic wavefunction $\Psi_g$ having a well-defined ground-state configuration (Supplementary Tables 1 and 2). Within the so-called crude adiabatic approximation[24] this ground-state configuration is defined at the equilibrium position $(Q_k)_0$ and is assumed to be independent of nuclear coordinates. For some modes, however, a proper description of $\Psi_g$ requires other configurations describing low-lying excited states to be incorporated. Its dynamical evolution along a relevant normal coordinate $Q_k$ is thus better described by the correlated configuration interaction of ground and excited configurations belonging, in this case, to a spin-doublet electronic manifold. Here we study the role of this electronic correlation on infrared mode activities of both spin-doublet monomers by applying three density-functional theory methods that gradually increase the level of electronic exchange-correlation (Fig. 3): B88→BLYP→ B3LYP. Also, we applied the mean-field method ROHF that does not include correlated dynamics.

**Table 1 | Infrared multiple-photon action bands of picene$^+$ and pentacene$^+$.**

| Infrared multiple-photon action | | | Deconvoluted fitted peak components | | | | | | | | B3LYP/6-311G(d,p) theory | | | | | $\delta(\tilde{v}_{exp}/\tilde{v}_{cal}-1)$ | $\beta/I_{cal}$ |
|---|---|---|---|---|---|---|---|---|---|---|---|---|---|---|---|---|---|
| Band | $\tilde{v}_{exp}$ | $\beta$ | Peak | $\tilde{v}_c$ | $\varepsilon$ | $w$ | $\varepsilon$ | $A$ | $\varepsilon$ | $h$ | $\tilde{v}_{cal}$ | $I_{cal}$ | $\nu_k(\Gamma)$ | $\tilde{v}_k$ | $I_k$ | | |
| *Picene$^+$* | | | | | | | | | | | | | | | | | |
| $a_r$ | - | - | 1 | 1,591 | 9.35 | 199 | 19.01 | 105 | 19.44 | 0.34 | 1573 | 0.31 | $\nu_{76}$ (b$_2$) | 1,577 | 250 | - | - |
| $a$ | 1,513 | 0.91 | 2 | 1,509 | 1.09 | 103 | 5.6 | 105 | 13.45 | 0.65 | 1529 | 0.94 | $\nu_{78}$ (b$_2$) | 1,528 | 1067 | −1.05 | 0.97 |
| $b_r$ | 1,335 | 0.75 | 3 | 1,347 | 1.11 | 63 | 4.35 | 34 | 4.21 | 0.34 | 1374 | 0.32 | $\nu_{83}$ (b$_2$) | 1,375 | 353 | −2.84 | 2.34 |
| $b$ | 1,282 | 0.93 | 4 | 1,282 | 0.82 | 92 | 5.55 | 98 | 8.44 | 0.68 | 1318 | 1 | $\nu_{84}$ (b$_2$) | 1,318 | 1407 | −2.73 | 0.93 |
| $b_l$ | 1,217 | 0.77 | 5 | 1,212 | 0.9 | 68 | 4.4 | 42 | 4.53 | 0.4 | 1232 | 0.4 | $\nu_{87}$ (b$_2$) | 1,232 | 497 | −1.22 | 1.93 |
| $c$ | 1,134 | 0.6 | 6 | 1,132 | 0.7 | 73 | 3.75 | 41 | 3.08 | 0.36 | 1131 | 0.23 | $\nu_{90}$ (b$_2$) | 1,133 | 190 | 0.27 | 2.61 |
| $d$ | 996 | 0.26 | 7 | 1,013 | 4.21 | 204 | 18.21 | 60 | 6.14 | 0.19 | - | - | $\nu_{93}$ (b$_2$) | 1,019 | 0.22 | - | - |
| $e$ | 823 | 0.23 | 8 | 823 | 1.81 | 67 | 7.25 | 16 | 1.66 | 0.15 | 830 | 0.076 | $\nu_{58}$ (b$_1$) | 818 | 64 | −0.84 | 3.03 |
| $f$ | 737 | 0.24 | 9 | 737 | 0.56 | 41 | 2 | 11 | 0.56 | 0.17 | 745 | 0.072 | $\nu_{59}$ (b$_1$) | 747 | 89 | −1.07 | 3.33 |
| *Pentacene$^+$* | | | | | | | | | | | | | | | | | |
| $a_r$ | 1,540 | 0.49 | 1 | 1,544 | 4.82 | 132 | 10.09 | 63 | 9.61 | 0.3 | 1521 | 0.36 | $\nu_{80}$ (b$_{2u}$) | 1,520 | 263 | 1.25 | 1.36 |
| $a$ | 1,458 | 0.65 | 2 | 1,458 | 2.1 | 102 | 13.2 | 64 | 12.91 | 0.4 | 1481 | 0.52 | $\nu_{81}$ (b$_{2u}$) | 1,481 | 418 | −1.55 | 1.25 |
| $b_r$ | 1,376 | 0.73 | 3 | 1,380 | 1.57 | 64 | 8.83 | 31 | 6.61 | 0.31 | 1373 | 0.9 | $\nu_{83}$ (b$_{2u}$) | 1,384 | 265 | 0.22 | 0.81 |
| $b$ | 1,319 | 0.94 | 4 | 1,317 | 0.83 | 78 | 2.23 | 91 | 3.74 | 0.74 | 1358 | 1 | $\nu_{85}$ (b$_{2u}$) | 1,355 | 684 | −2.87 | 0.94 |
| $c_r$ | 1,202 | 0.75 | 5 | 1,204 | 0.52 | 29 | 1.85 | 19 | 1.55 | 0.42 | 1221 | 0.22 | $\nu_{86}$ (b$_{2u}$) | 1,221 | 170 | −1.56 | 3.41 |
| $c$ | 1,171 | 0.9 | 6 | 1,170 | 0.37 | 36 | 1.78 | 39 | 2.16 | 0.7 | 1164 | 0.25 | $\nu_{87}$ (b$_{2u}$) | 1,164 | 185 | 0.6 | 3.6 |
| $c_l$ | 1,138 | 0.49 | 7 | 1,137 | 0.61 | 15 | 2.42 | 4 | 0.74 | 0.18 | 1141 | 0.15 | $\nu_{88}$ (b$_{2u}$) | 1,141 | 65 | −0.26 | 3.27 |
| $d$ | 1,076 | 0.39 | 8 | 1,077 | 1.95 | 31 | 5.85 | 12 | 1.95 | 0.25 | - | - | $\nu_{71}$ (b$_{1u}$) | 1,113 | 4 | - | - |
| $e_r$ | 968 | 0.34 | 9 | 974 | 6.63 | 78 | 22.09 | 24 | 7.79 | 0.19 | - | - | $\nu_{94}$ (b$_{3u}$) | 965 | 8 | - | - |
| $e$ | 910 | 0.53 | 10 | 908 | 2.2 | 59 | 6.9 | 39 | 6.2 | 0.42 | 926 | 0.076 | $\nu_{95}$ (b$_{3u}$) | 926 | 64 | −1.73 | 6.97 |
| $f$ | 738 | 0.52 | 11 | 738 | 0.56 | 18 | 1.24 | 13 | 0.66 | 0.46 | 743 | 0.103 | $\nu_{97}$ (b$_{3u}$) | 744 | 91 | −0.67 | 5.05 |

The D$_{2h}$ point-group symmetry of pentacene is conserved after ultraviolet photoionization, so pentacene$^+$ normal modes are classified in 8 symmetry representations ($\Gamma$) as $18a_g + 7b_{1g} + 9b_{2g} + 17b_{3g} + 8a_u + 17b_{1u} + 17b_{2u} + 9b_{3u}$. Likewise, the modes of C$_{2v}$-symmetry picene$^+$ are $35a_1 + 17a_2 + 16b_1 + 34b_2$. The column headings are action band label (Band), action band peak frequency ($\tilde{v}_{exp}$, cm$^{-1}$), normalized dissociation yield ($\beta$). The deconvoluted spectral band peak components (Peak) are obtained by fitting a Lorentzian function $y = y_o + (2A/\pi)(w/4(\tilde{v}-\tilde{v}_c)^2 + w^2)$ to action band features using a Levenberg–Marquart algorithm. The peak parameters with errors s.e. ($\varepsilon$, significance level 5%) are central frequency ($\tilde{v}_c$, cm$^{-1}$), full-width at half maximum ($\tilde{w}$), area ($A$) and height ($h$, $2A/\pi w$). The offsets ($y_o$) are 0.002 (picene$^+$) and 0.035 (pentacene$^+$). The statistics of the fits are $dof = 2,324$, $R^2$ (COD) = 0.98649, Reduced $\chi^2$ = 0.00108 (picene$^+$), and $dof = 2760$, $R^2$ (COD) = 0.96268, and Reduced $\chi^2$ = 0.00266 (pentacene$^+$). The theoretical quantum-chemical quantities are calculated band peak frequency ($\tilde{v}_{cal}$, cm$^{-1}$), calculated band relative intensity ($I_{cal}$), normal mode ($\nu_k$), normal mode frequency ($\tilde{v}_k$, cm$^{-1}$), and infrared mode oscillator-strength ($I_k$, km mol$^{-1}$). We report the band frequency shift deviations ($\delta$) relative to the calculation and experimental-to-theoretical relative intensity ratios ($\beta/I_{cal}$). A negative shift represents a band redshift. The absolute total shift obtained as the mean value of absolute shifts $|\delta|$ is 1.43 ± 0.9% for picene$^+$ and 1.19 ± 0.79% for pentacene$^+$.

The comparison between experimental and theoretical spectra show that the mean-field theory diverges more severely in picene$^+$, particularly in the C=C stretch high-energy region, suggesting the need for electron correlated dynamics to properly describe its modes. This suggests that picene$^+$ wavefunction readily interacts with certain excited electronic configurations during C=C stretch nuclear displacements, described in this case by b$_2$-symmetry normal modes, which is a clear sign of vibronic activity (see next section). The b$_2$-mode $\nu_{92}$ exhibits for instance an unphysical mean-field intensity of *ca.* 4,000 km mol$^{-1}$ (Fig. 3b) since the ROHF method is unable to simulate the effective intramolecular electric field responsible for the generated dipole derivative. However, the density functionals rectify the odd intensities as corroborated by the experimental spectrum. Despite not including explicit vibronic treatments, their better performance is explained by the fact that density functionals implement local and non-local exchange-correlation interactions based on the single-valued electronic density[42,43]. In this sense, including gradient-density corrections (which account for density fluctuations emerging from low-lying excited configurations) further rectify the C=C stretch band pattern in agreement with the measured relative strengths of picene$^+$ bands $a$ and $b$, for example, B88→B3LYP (Fig. 3c,e). For pentacene$^+$, the spectroscopic comparisons suggest an evolution of its wavefunction $\Psi_g$ along normal modes that roughly can be described by a single configuration; yet, some degree of correlated configuration interaction is required to reproduce the intensities of high-energy b$_{2u}$ modes.

For the out-of-plane C–H bending modes of the mid-energy region, all methods predict similar infrared activities except for some mode frequency shifts. The mean-field intensities of modes carrying $e$ and $f$ bands are 61 and 110 km mol$^{-1}$ for $\nu_{95}$ and $\nu_{97}$ in pentacene$^+$, and 68 and 112 km mol$^{-1}$ for $\nu_{58}$ and $\nu_{59}$ in picene$^+$. These intensities agree with B3LYP intensities (Table 1), showing that out-of-plane infrared-active modes are invariant to the level of theory.

**Vibronically driven π-fluxes.** The previous section shows that electron–electron correlated interactions are required for a proper description of dipole derivatives for C=C stretch b$_2$-modes of picene$^+$, and to a certain extent, also for C=C stretch b$_{2u}$-modes of pentacene$^+$. Such electronic correlation manifests itself in equation (1) as non-negligible vibronic matrix elements that represent a configuration-interaction expansion of excited-to-ground state mixing coefficients. Together with the electronic matrix term, they comprise the vibronic dipolar term enabling molecular π-electron fluxes. To elucidate the physics of dipolar π-fluxes we first analyse the symmetry properties of both matrix elements. In the analysis, we consider the modes $\nu_{84}$ and $\nu_{85}$ responsible for the action band $b$ of picene$^+$ and pentacene$^+$, respectively, since their largest infrared strengths (Table 1) suggest significant dipolar π-flux contributions to action spectra at comparable FEL excitation energies ($\Delta\tilde{v}_{exp}(b) = 2.8\%$). Figure 4 shows key π→π$^\star$ excitations in both systems for which electronic configurations are included in Supplementary Tables 1 and 2.

In view of the energy denominator in equation (1) one expects that the electronic wavefunction of picene$^+$ is most susceptible to mixing with the $\Psi_i(4a_2 \rightarrow 4b_1)$ excited configuration (Fig. 4a). On departure from its equilibrium geometry picene$^+$ thus needs to be described as a linear combination of ground and excited configurations $\Psi_g \approx \Psi_o(^2B_1) + c_i \, \Psi_i(4a_2 \rightarrow 4b_1)$, where the

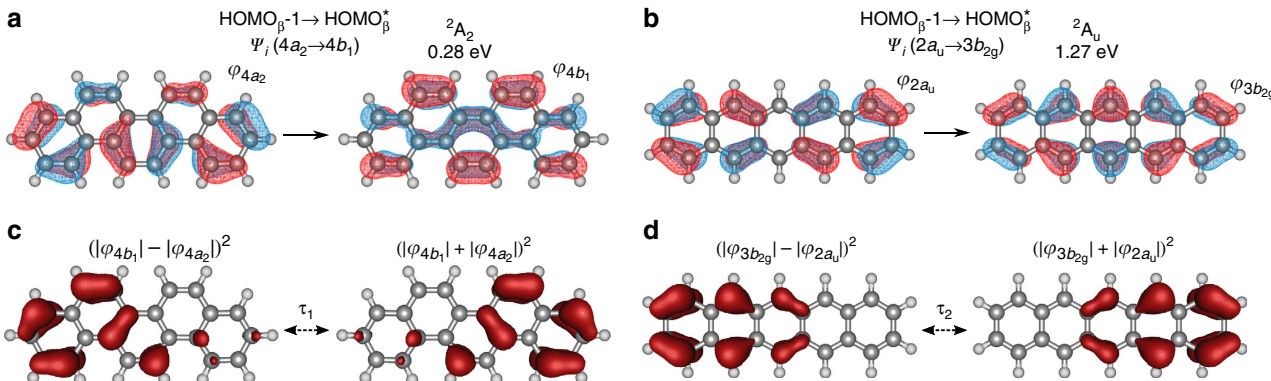

**Figure 4 | Electronic $\pi \to \pi^\star$ excitations and excited $\pi$-electron dipolar distributions.** (**a,b**) Vibronically active one-electron configuration excitations $\Psi_i(4a_2 \to 4b_1)$ and $\Psi_i(2a_u \to 3b_{2g})$ involving the spin-orbital $\pi$ wavefunctions $\varphi_{4a_2}$ and $\varphi_{4b_1}$ of picene$^+$ and spin-orbital $\pi$ wavefunctions $\varphi_{2a_u}$ and $\varphi_{3b_{2g}}$ of pentacene$^+$, respectively. The two electronic configurations describe excited $\pi$ states (Supplementary Tables 1 and 2) that mix into the ground state $\Psi_o$ (0 eV). The symmetry product rule $\Gamma(\Psi_o) \times \Gamma(\partial H/\partial Q) \times \Gamma(\Psi_i)$ determines the coupling excited states. For picene$^+$, the vibronic operator $\partial H/\partial Q$ transforms as $\nu_{84}$, hence $\Gamma(\partial H/\partial Q) = b_2(y)$, and $\Gamma(\Psi_o) = b_1$ which leads to excited-state wavefunctions $\Psi_i$ of $a_2$ symmetry. Similarly for pentacene$^+$, the symmetry of excited states promoting $\Pi$-fluxes along $\nu_{85}$ is $a_u$. (**c,d**) Probability spatial distributions of the excited $\pi$-electron density that oscillates between extreme values $(|\varphi_{4b_1}| - |\varphi_{4a_2}|)^2$ and $(|\varphi_{4b_1}| + |\varphi_{4a_2}|)^2$ in picene$^+$ and extreme values $(|\varphi_{3b_{2g}}| - |\varphi_{2a_u}|)^2$ and $(|\varphi_{3b_{2g}}| + |\varphi_{2a_u}|)^2$ in pentacene$^+$. The $\pi$ spin-orbital energy gaps (Fig. 1b) determine the oscillation periods ($\tau_1$, $\tau_2$) on the order of a few femtoseconds. The two-carbon and one-carbon density centers determine, respectively, the bonding and anti-bonding electronic characters of the excited spatial redistributions. Molecular structures and spin-orbital wavefunctions from B3LYP/6-311G** calculations.

vibronic mixing coefficient $c_i$ is proportional to $\langle \Psi_o | (\partial H / \partial Q_k)_0 | \Psi_i \rangle Q_k$. To visualize the implications of this mixing on the charge distribution dynamics, we consider in the following the limiting situation in which a one-on-one mixing occurs. During vibrational motion, the $\pi$-electron density then oscillates between the extreme spatial probability values $(|\varphi_{4b_1}| - |\varphi_{4a_2}|)^2$ and $(|\varphi_{4b_1}| + |\varphi_{4a_2}|)^2$, which are localized on the left and right hand parts of picene$^+$, respectively (Fig. 4c). Similarly during vibrational motion along mode $\nu_{85}$ the wavefunction of pentacene$^+$ needs to be described as $\Psi_g \approx \Psi_o$ ($^2B_{2g}$) $+ c_i \Psi_i(2a_u \to 3b_{2g})$ leading to a left-to-right redistribution of $\pi$-electron density generated by the mixed-in configuration $\Psi_i(2a_u \to 3b_{2g})$ (Fig. 4b,d). For both armchair- and zigzag-edge topologies it would thus appear that oscillating $\pi$-fluxes are generated.

To determine the $\pi$-flux strength we examine the vibronic coupling phases between the excited $\pi$-electron redistributions and the nuclear positions at the classical turning points (Fig. 5). We begin with picene$^+$ $-Q_{84}$ and $+Q_{84}$ positions, in which the amplified (or reduced) $\pi$-electron density in contracted (or stretched) bonds maintains in-phase or out-of-phase relationships with the limiting $\pi$-electron regions $(|\varphi_{4b_1}| - |\varphi_{4a_2}|)^2$ and $(|\varphi_{4b_1}| + |\varphi_{4a_2}|)^2$ (Fig. 5a,c). The phase relationships reflect the bond stabilizing ($+$) and bond destabilizing ($-$) character of the two-carbon density centers induced by mode $\nu_{84}$. With this terminology the emergent phases depicted in Fig. 5c are expressed as ($4+,0-|0+,4-$) for $-Q_{84}$ and ($4-,0+|0-,4+$) for $+Q_{84}$, where the vertical bar denotes the mirror molecular $xz$-plane. Evidently, at position $-Q_{84}$ the left side features an accumulation (scaling by a factor of 4) of bonding $\pi$-electron density relative to the right side, while at $+Q_{84}$ the $\pi$-density positive buildup occurs over the rings on the right side. Likewise for pentacene$^+$, Figure 5d shows the emergent phases during the vibronic mixing of $\Psi_i(2a_u \to 3b_{2g})$, such as ($2+, 2-|2+,2-$) at $-Q_{85}$ and ($2-,2+|2-,2+$) at $+Q_{85}$. In this case, the excited $\pi$-electron oscillation and geometrical changes are slightly unsynchronized in a way that it generates a $\pi$-density build-up (scaling by a factor of 2) with some neutralization at the center (note the relatively small inner-ring two-carbon centers located at opposite mirrored sides). This analysis indicates that

the $\pi$-flux strength in picene$^+$ is twice as much as in pentacene$^+$, which is in good agreement with the predicted infrared-strength ratio of $\nu_{84}$ (picene$^+$) over $\nu_{85}$ (pentacene$^+$) modes ($I_{84}/I_{85} = 2.06$, Table 1). We thus conclude that the dipolar source of these two C=C stretch modes is entirely due to the charge $\pi$-flux.

Experimentally, we observe in the action spectra a similar twofold $\pi$-flux enhancement once we calibrate the spectra that have been previously normalized on the intensity of band $b$ ($\beta_b$) with maximum absolute dissociation yield. To this end, we consider the intrinsic relation between $f$ and $b$ bands to be preserved in each system and then renormalize the spectra such that $\beta_f[\text{picene}^+] = \beta_f[\text{pentacene}^+] = (0.24/0.93)\beta_b[\text{picene}^+] = (0.52/0.94)\beta_b[\text{pentacene}^+]$ (Table 1). We have used the mid-energy band $f$ since in both systems this band is carried by intrinsically equivalent out-of-plane C–H bending modes. Solving for $\beta_b[\text{picene}^+]$ results in $2.15\beta_b[\text{pentacene}^+]$, which remarkably agrees with the strength ratio $I_{84}/I_{85}$, and thus, confirms the twofold increase of $\pi$-flux strength in picene$^+$.

**Degree of non-linearity in action spectra.** The above analysis shows that relative intensities of action bands $f$ and $b$ can be described under the linear-absorption harmonic approximation. However, we have mentioned that action spectra are inherently susceptible to nonlinearities as introduced by the multiple-photon dissociation (Fig. 2b,d). Moreover, relative intensities are based on dissociation yield functions that have been power-corrected and normalized, and these procedures could have brought an accidental agreement to the harmonic ratio $I_{84}/I_{85}$. Therefore, to estimate the extent of these effects, we compare spectra from control measurements at high-energy FEL macropulses with a spectra sample from the final average of Fig. 3. Note that the spectra retrieved from the control measurements are not part of the final average.

We first compare in Fig. 6 the $C_2H_2$ and $C_4H_4$ loss ion signals of picene$^+$ recorded at 0 and 3 dB FEL power levels (signal and band characterizations in Supplementary Tables 18–20). The signal ratio $C_4H_4$-loss/$C_2H_2$-loss is found approximately constant along FEL photon energy (Fig. 6), suggesting that $C_2H_2$ and $C_4H_4$ loss rates evolve in quasi-linear proportion to each other. On

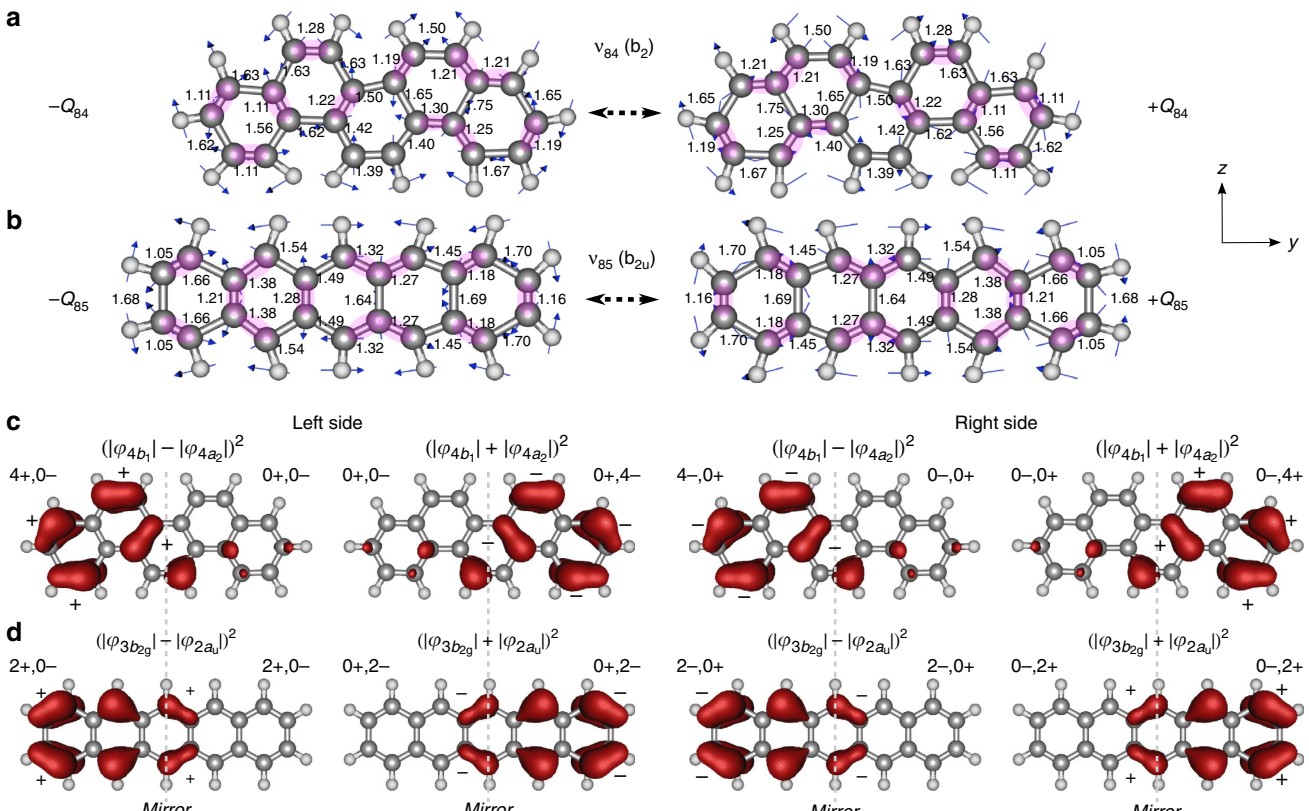

**Figure 5 | Vibronic C=C stretch antisymmetric modes and π-electron–mode coupling phases.** Displaced cationic structures linked by non-totally symmetric C=C stretch modes (**a**) $\nu_{84}$ (picene$^+$) and (**b**) $\nu_{85}$ (pentacene$^+$). The vibrational motion of these modes consists of in-plane C=C stretches with a perfect antisymmetric reflection relative to the molecular $xz$ plane (perpendicular to this page). The electronic density amplification in contracted bonds is coloured purple. Bond length values shown in Å. (**c,d**) Vibronic coupling phases between excited π-electron density redistributions and maximum displaced nuclear positions for picene$^+$ and pentacene$^+$ reveal the positive π-density build-up on left or right sides relative to the molecular $xz$ plane. The smaller-size signs depicted for the two-carbon density centers over the centrosymmetric inner ring in pentacene$^+$ illustrate a weak vibronic coupling as based on the relatively smaller C=C bond-length changes and reduced π-electron density (see text). Molecular structures and spin-orbital wavefunctions from B3LYP/6-311G** calculations.

attenuation (3 dB), the mean value of the hence (quasi-)constant ratio of 1.73 changes to 2.52 always in favour of $C_2H_2$ loss. It is clear that at reduced macropulse energies, the energized molecular ion population reaches a lower average internal energy ($E$). This leads to a lower dissociation rate and eventually shuts down dissociation into the $C_4H_4$-loss channel with higher activation energy (that is, the inequality $E - E_1 - E_{iso} < E_2$ is fulfilled, see Fig. 2d). At the lowest FEL photon energies (*ca.* $< 900$ cm$^{-1}$) the ratio deviates from linearity towards $C_2H_2$ loss. For instance at the peak of band $f$, the ratio drops from 2.37 to 3.28 (Supplementary Table 20). Here the production of $C_4H_4$-loss ions drops as much as 38% while for $C_2H_2$-loss it drops only 14%. Comparing with band $a$, $C_2H_2$ and $C_4H_4$ losses decline by 30% and 45%, respectively. Despite the uneven variations between channels at band $f$, Fig. 6 does suggest that the dissociation $f$ yield decreases linearly on attenuation, revealing the determining role of $C_2H_2$-loss in the action spectra.

Whereas multiple-photon excitation with lower-energy macropulses decreases the $C_2H_2$ and $C_4H_4$ loss ion productions at $f$, $e$ and $a$ bands (an exception is band $e$ $C_2H_2$-loss, which retains the 0 dB signal), the $C_2H_2$ loss increases 17% at band $b$. This 17% increment is close to the observed 14% decrement of $C_4H_4$ loss (also at band $b$) suggesting linearly inversed dissociation kinetics between channels. Specifically for pentacene$^+$, we observe that $C_4H_4$ loss marginally varies on FEL irradiation along the tuned range, which correlates with the relatively lower oven temperatures set for sublimation. This shows that most energized

pentacene$^+$ ions have an average internal energy just above the $C_2H_2$-loss activation energy (Fig. 2d). Because $C_2H_2$-loss steers the multiple-photon dissociation kinetics in both systems, their dissociation yields along excitation have a strong $C_2H_2$-loss signal component, from which nonlinear effects are recognized to be much smaller than the inherent differences between the two molecular systems.

**Power dependence of action band intensities.** To test whether nonlinear effects on band intensities are negligible, we retrieve the spectra from the ion signals of Fig. 6 (Fig. 7a,b; band decompositions in Supplementary Tables 21 and 22). The resultant yield functions $\beta^\star(\tilde{v})$ are not normalized nor corrected for FEL power variations (as a function of excitation energy) to perform absolute comparisons between band intensities at 0 and 3 dB power levels. Table 2 lists four bands with their frequency peak ($\tilde{v}_{exp}$), absolute yield ($\beta^\star$) and average power ($P$). The yield variation upon FEL attenuation is taken as a power-law function $\beta^\star(P) = CP^m$ ($C$ and $m$ are constant and exponent factors). As seen in Fig. 6 for band $f$, a twofold power reduction (corresponding to a macropulse energy change from 42 to 20.8 mJ) brings a linear twofold yield change, that is, $m = 1$. For bands $a$ and $e$ the variations are nearly linear ($m = 0.8$). For band $b$ we find a non-linear behaviour ($m = 0.1$) as expected from the reversed 17% increasing signal behaviour of $C_2H_2$-loss as we reduce the macropulse energy (Fig. 6).

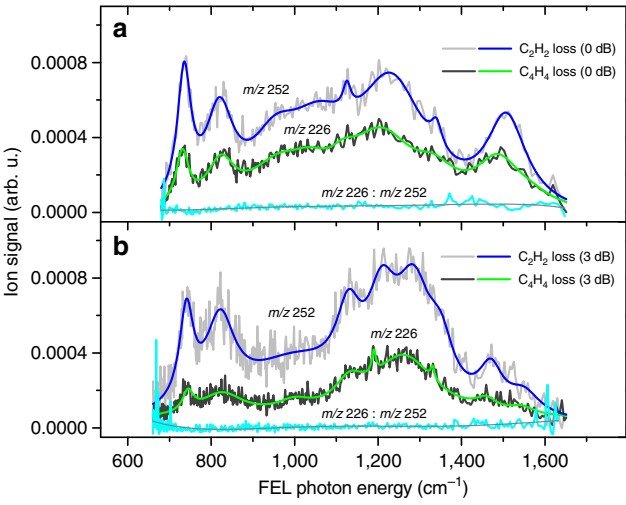

**Figure 6 | Individual ion signals of infrared multiple-photon dissociation channels of picene$^+$.** Recorded ion signals (grey) along FEL photon energy of $C_2H_2$ and $C_4H_4$ loss product ions before (**a**, 0 dB) and after (**b**, 3 dB) FEL power-level attenuation (corresponding to a macropulse energy reduction by a factor of 2). Each channel signal consists of two averaged FEL-scan data sets recorded at 0 dB (B0B1, sample from control measurements) and 3 dB (B2B3, sample from final spectra average) during a FEL beam-time session. The four signals are spectrally decomposed for which cumulative peak fit is depicted in blue ($C_2H_2$) and green ($C_4H_4$). The nearly constant ion signal ratio (cyan) between dissociation channels ($m/z$ 226 : $m/z$ 252)$^{-1}$ is 1.73 ± 0.55 (0 dB) and 2.52 ± 0.83 (3 dB). The outliers at the limits of the spectral range produce relatively large standard deviations, and result from dividing weak ion signals at the baseline level featuring a strong random noise factor. At the peak of band $f$, the signal values for $C_2H_2$ ($m/z$ 252) and $C_4H_4$ ($m/z$ 226) losses and parent ($m/z$ 278) (not shown) are 0.00034, 0.00081 and 0.070 at 0 dB, and 0.00021, 0.00069 and 0.099 at 3 dB (Supplementary Tables 18,19). Thus, the resultant dissociation yields are 0.0161 (0 dB) and 0.0090 (3 dB), which represent a 44% yield reduction on attenuation. Note that contrary to the band yields from partial spectra (Fig. 7) in which we usually integrate first the two product ion signals from the same scan measurement, here the band yields are obtained by integrating the same product ion from two scan measurements which then have somewhat different baselines and signal-to-noise levels. This brings an accuracy difference of 6% in comparison with the former procedure from which dissociation yields feature a 50% reduction on attenuation (see Table 2).

Comparing the absolute band yields of picene$^+$ with those of pentacene$^+$ (Fig. 7c, Supplementary Table 23) measured at the lower FEL power values reveals a reasonable agreement (compare Fig. 7b,c). The pentacene$^+$ $f$ yield intensity (0.00894) recorded at 26 mJ (260 mW) linearly scales to 0.0072 at 20.8 mJ in fair agreement with picene$^+$ $f$ yield 0.0077 recorded at 20.8 mJ (Table 2). This resemblance confirms the equal theoretical infrared strengths of their photoexcited modes ($\nu_{59}$ in picene$^+$ and $\nu_{97}$ in pentacene$^+$). For band $e$, we expect some deviation given the difference in excitation energies between picene$^+$ and pentacene$^+$ for this band (frequency deviation of $\Delta\tilde{\nu}_{exp}(e) = 10\%$ from Table 1), which suggests somewhat different anharmonic couplings. Indeed, at 19.3 mJ the pentacene$^+$ $e$ yield (measured at 23.9 mJ) scales to 0.0060 whereas the picene$^+$ $e$ yield is 0.0071 (Table 2).

For band $b$, the similar excitation energies required in both parent ions ($\Delta\tilde{\nu}_{exp}(b) = 2.8\%$) suggest that absolute yields could be comparable at the lower FEL power values (Fig. 7b,c). The $b$ yields are 0.0125 and 0.0153 recorded at 17.8 and 9.9 mJ for pentacene$^+$ and picene$^+$, respectively. For pentacene$^+$, the $b$

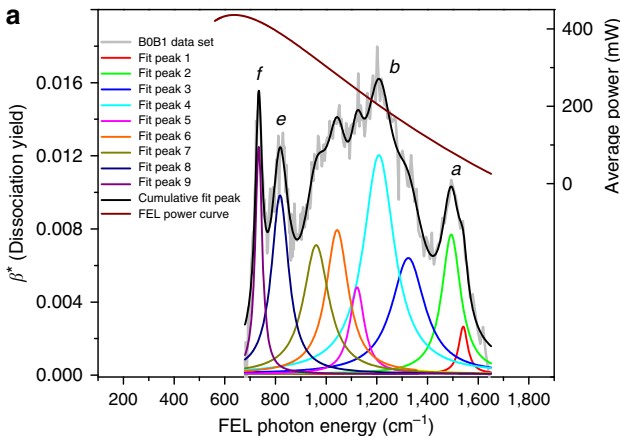

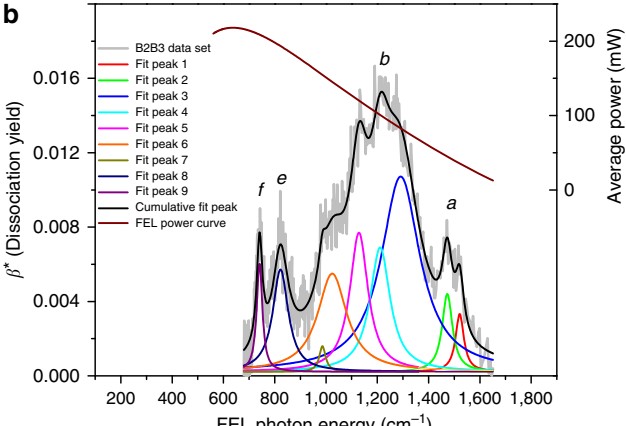

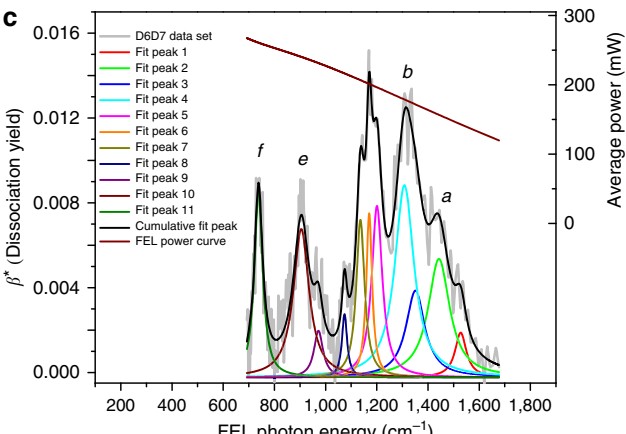

**Figure 7 | Partial action spectra without FEL power curve correction.** Picene$^+$ spectra (non-normalized) are retrieved from data sets consisting of two averaged FEL spectral scans along the depicted range of parent and product ions recorded without (**a**, 0 dB; B0B1), and with (**b**, 3 dB; B2B3) FEL power level attenuation. The B0B1 power curve is the same as the B2B3 power curve but as twice as high. The B0B1 data set is not part of the final picene$^+$ spectrum of Fig. 3a. The pentacene$^+$ spectrum (non-normalized) is retrieved from a data set (D6D7) of two averaged FEL spectral scans of parent and product ions (**c**). The FEL power curves are obtained from a polynomial fit to power-meter readings at the FEL output. The spectral band deconvolutions are summarized in Supplementary Tables 21–23. These fitted peaks differ from those reported in Fig. 3 since they belong to partial spectra featuring lower signal-to-noise levels.

**Table 2 | Absolute dissociation yield band intensities versus FEL average power.**

| Infrared band | $\tilde{v}_{exp}$ (cm$^{-1}$) | $P$ (mW) | $\beta^\star$ | $\Delta\tilde{v}_{exp}$ (%) | $P_{0\,dB}/P_{3\,dB}$ | $\beta^\star_{0\,dB}/\beta^\star_{3\,dB}$ | $m$ | $C$ |
|---|---|---|---|---|---|---|---|---|
| a | 1,489 | 43 | 0.0063 | 0.20 | 1.91 | 1.64 | 0.8 | 0.0004 |
|   | 1,492 | 82 | 0.0103 | | | | | |
| b | 1,219 | 99 | 0.0153 | 0.99 | 2.01 | 1.06 | 0.1 | 0.0103 |
|   | 1,207 | 199 | 0.0162 | | | | | |
| e | 824 | 193 | 0.0071 | 0.73 | 2.01 | 1.76 | 0.8 | 1E − 04 |
|   | 818 | 388 | 0.0125 | | | | | |
| f | 741 | 208 | 0.0077 | 0.81 | 2.02 | 2.02 | 1.0 | 4E − 05 |
|   | 735 | 420 | 0.0155 | | | | | |

Infrared band characterizations of action spectra of Fig. 7a,b. The column headings are action band label (Infrared band), action band peak frequency ($\tilde{v}_{exp}$), FEL average power ($P$), or macropulse energy ($P$/10 Hz, in mJ), absolute dissociation yield intensity ($\beta^\star$), band peak frequency shift deviation ($\Delta\tilde{v}_{exp}$), FEL average power ratio ($P_{0\,dB}/P_{3\,dB}$), yield ratio ($\beta^\star_{0\,dB}/\beta^\star_{3\,dB}$). The parameters of a fitted absolute-yield intensity power-law $\beta^\star(P) = CP^m$ are exponent ($m$) and constant ($C$) factors. Given the large frequency deviation between high- and low-power band a peaks ($\Delta\tilde{v}_{exp} > 1\%$), the lower-power band characterization is done at $P_{0\,dB}/P_{3\,dB} \approx 2$ (1.91). The a band $\beta^\star$ value thus differs from the one reported in Supplementary Table 22.

yield at 9.9 mJ linearly scales to 0.00695. This lower pentacene$^+$ intensity does reflect the theoretical strength of the photoexcited mode ($v_{85}$) carrying this band relative to the one of picene$^+$ ($v_{84}$) that is as twice as intense (that is, $I_{84}/I_{85} = 2.06$). Indeed, the ratio between absolute yields at 9.9 mJ is 2.2, which represents an accuracy error relative to the calculated $I_{84}/I_{85}$ ratio of 6.8%.

Finally, note that the random noise in band b intensities leads to standard deviation percentages of about ±6% (picene$^+$) and ±11% (pentacene$^+$) (Supplementary Fig. 2). These deviations are the largest attained in our experiments since they correspond to a smaller representative data sample of spectra for which noise component is evidently higher than in the full data sample average of Fig. 3. The random noise influences the accuracy of the spectral deconvolution fittings characterizing the experimental band intensities. Let us now use these deviations to investigate the ratio error in the case of the absolute intensities $\beta_b^\star$ reported above. The pentacene$^+$ $\beta_b^\star$ intensity is $0.0125 \pm 0.00138$, and the lower and upper $\beta_b^\star$ bounds linearly scaled to 9.9 mJ are 0.0062 and 0.0077 leading to ratio bounds relative to picene$^+$ ($\beta_b^\star = 0.0153$) of 1.98 and 2.47. For picene$^+$, we determine the ratio bounds relative to pentacene$^+$ at 17.8 mJ ($\beta_b^\star = 0.0125$) as 2.07 and 2.33. We thus establish that even at these limits the statistical error in our measurements justifies the band b ratio as also confirmed by its agreement with the theoretical value.

## Discussion

We have deduced the relative strength between vibronic $\pi$-flux contributions to modes $v_{84}$ and $v_{85}$ generating the action band b of picene$^+$ and pentacene$^+$, respectively. Within our FEL macropulse settings, the picene$^+$-to-pentacene$^+$ b band ratio is satisfactorily explained under the harmonic approximation which predicts an intensity ratio $I_{84}/I_{85}$ of 2.06 (at B3LYP/6-311G** level of theory). This means that a harmonic Hamiltonian can describe the molecular potential along action b band vibrations. One last question relates to the effects of spectral convolution on band intensities. On the basis of the heights ($h$) of deconvoluted peaks for f and b bands, we estimate an experimental band ratio of 2.41 (where we subtracted the 0.035 offset for pentacene$^+$ peaks, see Table 1). This value deviates by 12% from the first value obtained from relative intensities in Fig. 3. The better agreement in the first estimation could be due to an error cancelation on averaging over the full data set, but also, it could reflect the absolute intensity difference of picene$^+$ (0.0077) and pentacene$^+$ (0.0072) f bands. Nonetheless, the fair agreement between action and harmonic band intensity ratios ratifies the twofold $\pi$-flux increase in picene$^+$ compared with pentacene$^+$.

From our studies a picture emerges in which picene is able to generate a higher degree of dipolar charge separation along its monomeric structure than pentacene. We have argued that this is a direct consequence of the spatial dynamical evolution (via the adiabatic vibronic operator $(\partial H/\partial Q)_0$) of mixed armchair-edge type electronic wavefunctions typical of angular-oriented aromatic structures. These armchair-edge wavefunctions are intrinsically strongly correlated. Thus, in the case of superconducting molecular crystals based on picene and related monomers[9,12], this evidence implies that both electron–phonon and electron–electron interactions could be at the origin of electron pairing. This work allows us to also envision the exploration of picene-like motifs in heterojunctions[44] or supramolecular nano-assemblies[45] built in devices, whose operation is triggered by photo-induced charge separation. Finally, note in equation (1) that larger vibronic couplings along modes of armchair-type monomers relative to zigzag-type monomers may not always translate in larger dipolar charge-flux contributions to the infrared spectra in the former class, since the vibronic coupling strength is weighted by the electronic transition matrix element. In conclusion, our results show that infrared multiple-photon action spectroscopy can deliver significant information on intramolecular charge dynamics when applied to charged molecular species with similar dissociation kinetics and vibrational resonances.

## Methods

**Free-Electron Laser for Infrared eXperiments FELIX.** In our experiments, FELIX delivered typically 7 μs long macropulses every 100 ms with transform-limited bandwidth of about 1% of the central $\lambda$ tuned in steps of 0.02 or 0.04 μm. Each macropulse consisted of 1 ps long micropulses at 1 GHz. A typical average energy at 13 μm was 42 mJ per macropulse, which delivers a fluence of 5.3 J cm$^{-2}$ in the center of the ion trap on a spot of 1-mm diameter. Nowadays, FELIX is located at Radboud University in Nijmegen (The Netherlands).

**Infrared multiple-photon dissociation action spectroscopy.** Action spectra are recorded with a Paul-type ion trap mass-spectrometer (Jordan TOF Products, Inc.) attached to a FELIX beamline[20,39]. The ion trap built into a high vacuum chamber is made of a toroidal inner ring electrode of 2 cm inner diameter interposed between two hyperbolic endcap electrodes. The trap is biased at +1,000 VDC, setting a potential difference relative to a 60-cm length time-of-flight (TOF) mass spectrometer used for mass-to-charge ($m/z$) ion detection. Cations ($m/z$ 278) of pentacene and picene are produced by 193 nm ultraviolet photoionization of gasphase neutral molecules effused to the inner trap volume upon sublimation of solid samples (99.9% picene, TCI Europe; 99% pentacene, Sigma-Aldrich) with a built-in oven at temperatures as high as 200 °C. The ultraviolet source is a 5-ns pulsed excimer laser (PSX-501 Neweks Ltd.) adjusted to a typical energy of 1.6 mJ per pulse and power density in the trap of $3 \times 10^5$ W cm$^{-2}$. For TOF mass analysis, axial extraction is achieved by switching off the RF voltage while applying a −250 VDC pulse to the endcap (with a 3 mm hole) closest to the TOF-tube. Ultraviolet photo-induced product ions below $m/z$ 278 are ejected before FEL irradiation by a brief (2 ms) RF amplitude increase. A few milliseconds after isolation of an ensemble of picene$^+$ (or pentacene$^+$) ions, FEL on-resonance irradiation at fundamental vibrational transitions induces multiple-photon dissociation. Two spherical mirrors (gold coated) are used to enhance the FEL fluence. A FEL-triggered delay generator (SRS-DG520) controls the 10 Hz experimental sequence. The recorded ion signals of parent and multiple-photon product ions are amplified and digitized (Acquiris). The pressure values in the high vacuum chamber were typically in the $10^{-6}$–$10^{-7}$ mbar range. We estimate the dissociation yield, $\beta(\lambda)$ or

$\beta(\bar{v})$, to be proportional to the absorption cross-section, that is, the ratio of total product ions over the sum of total product and parent ions as function of FEL photon energy wavelength. We control the infrared fluence by selecting a FEL power level (dB; via a calibrated set of wire-mesh attenuators). The FEL wavelength is calibrated using a grating spectrometer and the dissociation yields are linearly corrected for power variations across the scan range. The final action spectra are obtained from averaging about 10–12 dissociation yield spectral functions $\beta$ including the low spectral range. The spectral deconvolution curve fits of action spectra into Lorentzian peak components and further data processing are carried out in Origin (OriginLab, Northampton).

**Quantum chemical calculations.** We obtain optimized electronic structures and harmonic vibrational mode frequencies using *ab initio* Hartree–Fock theory as the spin-restricted open-shell ROHF method[46], and the local spin-density approximation (LSDA) plus a variety of exchange-correlation gradient-corrected formalisms. These are the LSDA gradient-corrected exchange-only Becke-1988 functional[43] (B88), the Becke-1988 method plus the gradient-corrected correlation LYP functional[42] (BLYP), and the hybrid 3-parameter functional that includes generalized exchange-correlation gradient corrections and some degree of exact Hartree–Fock exchange energy[47] (B3LYP). The atomic-orbital basis set used is a split-valence triple-$\zeta$ Gaussian-type 6-311G with $d$ and $p$ polarization functions. The excited-state calculations were done at the B3LYP level after transposing the relevant spin-orbitals involved in the excitations. To reduce computational cost we used the smaller set 6-31G($d$,$p$). All reported harmonic spectra were generated by convoluting normal modes with a Lorentzian $30\,cm^{-1}$ bandwidth profile with frequencies scaled by 0.97 to account for basis-set truncation. The excited spatial distribution probabilities were obtained from Kohn–Sham spin-orbital $\pi$ wavefunctions $\varphi$. Since $\text{Re}\{\varphi\} = \varphi$ the complex conjugated $\varphi^*$ is $\varphi$ and the modulus $|\varphi| = (\varphi^*\varphi)^{1/2} = (\varphi\varphi)^{1/2} = \varphi$. We performed all quantum-chemical calculations using Gaussian 09 (Frisch, M.J. *et al.* Gaussian 09, Revision A.02., 2009) at the SurfSARA computing facility in Amsterdam.

**Data availability.** The data supporting the findings of this study are available within the article, Supplementary Information, and if applicable, from the corresponding author on request.

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

## Acknowledgements

We gratefully acknowledge the excellent support of A.F.G. van der Meer as well as Michel Riet, Rene van Buuren, Jules van Leusden, Giel Berden, and Joost Bakker at the FELIX facility. This work was part of the research program of the Stichting voor Fundamenteel Onderzoek der Materie, which was financially supported by the Nederlandse Organisatie voor Wetenschappelijk Onderzoek. Computing time at the SurfSARA supercomputer center was kindly provided by NWO-EW under grant MP-264.

## Author contributions

B.R. supervised, maintained and operated FELIX, and assisted in FELIX-ion-trap experiments. W.J.B. performed the calculations and analysis. J.O. supervised the research, performed analysis, and assisted in FELIX-ion-trap experiments. H.A.G. conceived the study, performed the FELIX-ion-trap experiments, calculations and analysis, and wrote the manuscript with inputs of W.J.B. and J.O.

## Additional information

**Competing financial interests:** The authors declare no competing financial interests.

**How to cite this article**: Álvaro Galué, H. *et al.* Electron-flux infrared response to varying π-bond topology in charged aromatic monomers. *Nat. Commun.* 7:12633 doi: 10.1038/ncomms12633 (2016).

