## [Peer Review File · Nature Communications]

Reviewers' comments:

Reviewer #1 (Remarks to the Author):

To investigate the role of topology of pi-electron network on the vibronic effects in electric conductivity, the authors observed picene and pentacene cations with infrared multiple-photon absorption spectroscopy. Their findings are important in the field of organic electronics. Therefore, the present manuscript is publishable after revision.

The authors discussed on the electronic correlation effect by comparing ROHF and unrestricted DFT calculations. It has been reported that broken-symmetry of spacial orbitals also plays an important role in vibronic couplings[J. Phys. Chem. A 112, 758 (2008).]. Therefore, to demonstrate the electronic correlation effect separated from the broken-symmetry effect, they should employ restricted DFT calculation, or UHF method in place of the ROHF calculation.

Reviewer #2 (Remarks to the Author):

This manuscript reports electronic structures, harmonic analyses of IR action spectra, electronic correlation and vibronically driven n -fluxes for picene and pentacene. The important point derived from this study is that picene can generate a higher degree of dipolar charge distribution along its monomeric structure than pentacene. The authors argued that the above result originated from the spatial dynamic evolution of mixed armchair-edge type electronic wavefunctions in picene. Consequently, the electron-phonon and electron-electron interaction become stronger to strengthen the electron pairing. This scenario may be reasonable to explain the superconductivity of picene. Furthermore, it is interesting to find the strong electron-phonon and electron-electron interactions characteristic of armchair-edge type molecule.

This paper is well written, and it may be important for the researchers of the confined field (researches of organic superconductors and carbon-based superconductors). However, it may not attract much attention from a wide variety of readers in physics and chemistry. I recommend this paper will be submitted to other general journals (Scientific Reports, Physical Review B).

Reviewer #3 (Remarks to the Author):

A) The manuscript by Galue, Ommens, Buma and Bredlich discuss the enhancement of molecular vibration intensities via coupling with delocalized pi-electrons, which is also known to be important in charge conductivity. Two benchmark molecules, picene and pentacene, are considered in order to draw conclusions about the role of the edge topology of their respective aromatic motifs on this flux mode enhancement. It is hypothesized that the stronger dipolar charge-pi flux for picene is consistent with the electronic properties of "armchair"-type molecules, compared with the "zigzag"-type molecule pentacene.

B) In general, the topic is innovative and the potential implications from this research are far-reaching,

making me believe that the work is in principle suitable for the broad readership of Nature.

C) It appears to me that the authors are over-interpreting (and/or too selectively interpreting) their data, and that the conclusions thus are mostly tentative. The seminal point that the manuscript discusses applies to the comparison of the experimental and computed IR intensity ratios of the 84 mode for picene vs. the 85 mode for pentacene. The enhancement factor of 2 (predicted by B3LYP) is also seen for the experimental bands, labeled "b", provided that those intensities are normalized by the band "f". There are a number of complications with this analysis:

First and foremost, IRMPD is based on the absorption of multiple photons, and is thus inherently non-linear in its behavior. Exceptional care must therefore be taken when using IRMPD intensities for the purpose of quantification. I would suggest that the data at all relevant peak positions, a-g, be reproduced at different laser powers. If it turns out that the relative band intensities are not affected by laser power, a much stronger case could be made for the hypothesis that IRMPD intensities are relative to the normal mode oscillator strengths.

It also seems to me that the "b" band in Fig 3f is close to saturated, which would make a quantification impossible. Lower laser power would also address this problem.

Finally, the predicted band intensities line up poorly with the experimental IRMPD spectrum. The "f" bands in particular seem to be vastly underestimated. This does not instill confidence in employing experimental IRMPD intensities for this purpose.

I also find the spectral interpretation fraught with some difficulties. The "b" band for pentacene is assumed to be exclusively assigned to mode 85, when in fact this is not what theory predicts - bands 84 and 83 likely also contribute to this feature. For the 84 mode for picene, the situation is arguably clearer. Why are not other modes (81 and 80 for pentacene, 78 for picene included in the analysis)? Finally, there is the question on how the IRMPD "intensities" should be evaluated. Do all of the bands have the bandwidths? If not, one would have to integrate bands, rather than take the maximum intensities.

D) Experimental uncertainties are not discussed appropriately. A plot of the exptl band ratios vs. laser power (as discussed in C) in fact could yield an approximation of the uncertainty in this analysis (i.e., a ratio of 2 +/- ??). Since a ratio of 2 is rather small, do the error bars justify the conclusion?

E) I cannot agree with the authors' conclusions based on their methodology and interpretation.

F) As largely already discussed in C, the linearity of IRMPD in these experiments needs to be tested rigorously. Also, more bands should be included in the theoretical analysis in order to evaluate this hypothesis.

G) Literature seems appropriate

H) The manuscript is well written

REVIEWERS' COMMENTS:

Reviewer #3 (Remarks to the Author):

My main previous criticism was aimed at section C (Data and methodology):

The authors have responded to these criticisms by including power-dependent IRMPD data, which in fact shows that the IRMPD response is close to linear. In addition, they have employed a more rigorous data interpretation approach, by deconvoluting the IRMPD bands. Crucially, this has enabled them to estimate an experimental error.

As my main concerns have been addressed, I am content with the current version of the manuscript, and thus recommend publication.

Please find attached a revised version of our manuscript in which we have taken into account the points raised by the three referees. In the following we will address their comments and how we have revised our manuscript accordingly.

Reviewer 1

The reviewer is overall positive, but indicates that symmetry breaking could play an important role in the theoretical analysis of vibronic couplings. The reference that is made to the J. Phys. Chem. A 112, 758 (2008) article is in our opinion not correct since this article does not discuss the role of symmetry breaking. Most probably, the reviewer intended to refer to the J. Chem. Phys. 124, 024314 (2006) article by the same group that discusses this point in detail. Our studies aim to show effects of electron correlation in picene and pentacene radical cations. We do this by comparing experimental spectra with theoretically predicted spectra with different levels of electron exchange-correlation taken into account. Thus for this basic demonstration, a detailed analysis of components contributing to specific vibronic couplings is not required (although we do agree with the reviewer that if we would want to do so, we should indeed make an extensive analysis of the role of symmetry breaking).

Reviewer 2

The reviewer recognizes the importance of our results, but is not entirely convinced that they will attract the attention of the broad scientific community. The other two reviewers, on the other hand, state that the implication of our work are far-reaching and will be of interest to a broad readership. We therefore consider this part of the conclusion of the reviewer as a non-issue.

Reviewer 3

The reviewer agrees that our results provide a significant breakthrough in understanding the electronic properties of technologically relevant organic molecules but is concerned with a number of aspects of our studies that primarily relate to the application and interpretation of IRMPD methodologies. We are grateful to the reviewer for his/her constructive comments that have led us to revisit and extend our analyses of experimental results. On the basis of these results we have now made major changes to the manuscript. In the following we will address the points raised by the referee in more detail and indicate the pertaining changes that we have made.

C)

(i) The reviewer notices that IR multiple-photon dissociation spectra can exhibit notable nonlinear effects. In our original manuscript it was implicit that these nonlinearities are much smaller than the inherent differences between the two molecules. In the revised manuscript we have expanded considerably on this issue and show in far more detail that action band intensities can be safely used in our analyses (see new sections “**Degree of non-linearity in action spectra**” and “**Power dependence of action band intensities**”).

(ii) The reviewer suspects that band 'b' is nearly saturated. However, bands 'b' (and 'f') were recorded with relatively low laser (FEL) power and nonlinearities are negligible as verified by the linear absorption comparisons with pentacene⁺ bands (Fig. 7).

(iii) The reviewer expresses his/her concerns with our spectral interpretation. However, the reviewer's conclusion that we assign the pentacene⁺ band 'b' exclusively to ν_{85} is not correct; already in the original manuscript we mentioned that this band includes broadening contributions from ν_{84} and ν_{83} . In the revised manuscript we take into account these contributions via a spectral band deconvolution procedure and show that they do not affect this study's conclusion (see below v).

(iv) The reviewer raises the question as to why other modes (ν_{81} and ν_{80} for pentacene, and ν_{78} for picene) have not been included in the analysis. The indicated pentacene⁺ modes have moderate IR strengths and their IRMPD bands are consequently overshadowed by the dominant IRMPD band of mode ν_{85} . The dominant picene⁺ ν_{78} mode can in principle be used but has no counterpart in pentacene⁺.

(v) The reviewer addresses how IRMPD intensities should be evaluated, suggesting the use of band areas rather than maximum intensities. In the revised manuscript we have now reported on spectral band deconvolutions of IRMPD spectra based on a Levenberg-Marquardt algorithm. Using the decomposed peaks (see Table 1), we estimate a picene⁺-to-pentacene⁺ band 'b' ratio of 2.41 which deviates by 12% from the initial estimate of 2.15. We discuss in the revision possible reasons for this deviation, although the value clearly still remains in good agreement with the theoretical value.

D)

The reviewer correctly mentions the lack of an appropriate analysis of the experimental uncertainties. Such an analysis is in the revised manuscript reported for a data sample set in Supplementary Figure 2. Also, the new section "Power dependence of action band intensities" analyzes the band intensity variations with laser (FEL) power. From these analyses it becomes clear that the experimental error indeed justifies our conclusions.

F)

The reviewer reiterates his/her concerns on the linearity of the IRMPD spectra, and suggests the use of more bands for the analysis. Above, we have shown that we fully agree with the concerns of the reviewer, but that in the present case these concerns have been taken care of. Concerning the use of more bands, it is important to notice that not all the measured bands provide vibronic charge-flux information. Modes 85 in pentacene⁺ and 84 in picene⁺ are in this respect the best reporters and have therefore been focused on in the present study.

We hope that with these changes the manuscript will be acceptable for publication in Nature Communications.

Sincerely yours,

Hector Alvaro Galué,
on behalf of all authors